# Mechanism for inverted-repeat recombination induced by a replication fork barrier

Léa Marie [1] & Lorraine S. Symington [1,2 ✉]

Replication stress and abundant repetitive sequences have emerged as primary conditions underlying genomic instability in eukaryotes. To gain insight into the mechanism of recombination between repeated sequences in the context of replication stress, we used a prokaryotic Tus/*Ter* barrier designed to induce transient replication fork stalling near inverted repeats in the budding yeast genome. Our study reveals that the replication fork block stimulates a unique recombination pathway dependent on Rad51 strand invasion and Rad52-Rad59 strand annealing activities, Mph1/Rad5 fork remodelers, Mre11/Exo1/Dna2 resection machineries, Rad1-Rad10 nuclease and DNA polymerase δ. Furthermore, we show recombination at stalled replication forks is limited by the Srs2 helicase and Mus81-Mms4/Yen1 nucleases. Physical analysis of the replication-associated recombinants revealed that half are associated with an inversion of sequence between the repeats. Based on our extensive genetic characterization, we propose a model for recombination of closely linked repeats that can robustly generate chromosome rearrangements.

[1] Department of Microbiology & Immunology, Columbia University Irving Medical Center, New York, NY 10032, USA. [2] Department of Genetics & Development, Columbia University Irving Medical Center, New York, NY 10032, USA. ✉email: lss5@cumc.columbia.edu

Maintaining genome integrity is essential for the accurate transmission of genetic information and cell survival. Replication stress has emerged as a major driver of genomic instability in normal and cancer cells. Replication forks become stressed as a result of DNA lesions, spontaneous formation of secondary structures, RNA–DNA hybrids, protein–DNA complexes, activation of oncogenes, or depletion of nucleotides[1]. These obstacles to the progression of replication can cause forks to slow down, stall and collapse. Consequently, multiple mechanisms have evolved to handle perturbed replication forks to ensure genome stability[2].

In eukaryotes, the presence of multiple replication origins, including dormant origins that are fired in response to replication stress, is one-way to ensure complete genome duplication[3]. Alternatively, the obstacle can be bypassed by translesion polymerases or by template switching. The latter is a strand exchange reaction mediated by homologous recombination (HR) proteins, consisting of annealing a nascent strand to its undamaged sister chromatid to template new DNA synthesis[4]. In recent years, replication fork reversal has also emerged as a central remodeling process in the recovery of replication in both eukaryotes and bacteria[5]. This process allows stalled replication forks to reverse their progression through the unwinding and annealing of the two nascent strands concomitant with reannealing of the parental duplex DNA, resulting in the formation of a four-way-junction, sometimes called a chicken-foot structure. Consequently, the lesion can be bypassed by extension of the leading strand using the lagging strand as a template followed by branch migration of the reversed structure. Alternatively, the extruded nascent strands can undergo HR-dependent invasion of the homologous sequence in the reformed parental dsDNA, resulting in the formation of a D-loop to restart replication. In bacteria, the replisome is reassembled on the D-loop structure[6], whereas in eukaryotes DNA synthesis within the D-loop can extend to the telomere or be terminated by a converging replication fork[7]. In addition, relocation of a lesion back into the parental duplex could facilitate repair by the excision repair pathways[8].

Thus, along with its critical role in DNA repair and segregation of chromosome homologs during meiosis, HR is involved in multiple replication restart mechanisms, which contribute to the preservation of genome integrity. However, HR can also be a source of instability as it occasionally occurs between chromosome homologs in diploid mitotic cells, resulting in loss of heterozygosity. Moreover, non-allelic HR (NAHR) between dispersed repeats can cause genome rearrangements[9–12]. A significant factor underlying chromosome rearrangements is the abundance of repeated sequences in eukaryotic genomes. Approximately 45% of the human genome is composed of repetitive sequences including transposon-derived repeats, processed pseudogenes, simple sequence repeats, tandemly repeated sequences, and low-copy repeats distributed across all chromosomes[13,14]. NAHR between repeated sequences can lead to deletions, duplications, inversions, or translocations[15,16]. Consequently, NAHR has been associated with many genomic disorders[17,18] and is a major contributor to copy-number variation in humans.

It is well established that rearrangements due to NAHR can result from the repair of double-strand breaks (DSBs)[19,20]. However, studies in yeast, human, and bacteria have shown that such genomic alterations can also arise during replication[12,21–23]. Notably, studies in Schizosaccharomyces pombe have shown that a protein-induced, site-specific replication fork barrier can cause a high frequency of genomic rearrangements in the absence of a long-lived DSB intermediate[22,24], consistent with the idea that replication stress contributes to NAHR. Elucidating the molecular mechanisms of NAHR occurring during the processing and restart of stressed replication forks remains crucial to understanding how genome rearrangements occur.

In Saccharomyces cerevisiae, spontaneous HR between repeated sequences shows different genetic requirements depending on the genomic location of the repeats. Inter-chromosomal recombination is generally Rad51-dependent, whereas recombination between tandem direct repeats can occur by Rad51-independent single-strand annealing (SSA)[25]. It has been shown that repeats in inverted orientation can spontaneously recombine by Rad51-dependent and Rad51-independent mechanisms[26], and these two pathways generate different recombination products. Rad51-mediated recombination results in gene conversion, which maintains the intervening sequence in the original configuration, whereas Rad51-independent recombination leads to an inversion of the intervening DNA. The inversion events require Rad52 and Rad59[27], which are known to catalyze annealing of RPA-coated single-stranded DNA (ssDNA) in vitro, and are required for SSA in vivo. Because DSB-induced recombination between inverted repeats is dependent on Rad51[28], it was proposed that the spontaneous Rad51-independent inversions could be the result of annealing between exposed ssDNA at stressed replication forks[29].

To elucidate the mechanism of NAHR between inverted repeats in the context of replication stress, we investigated the role of a protein-induced replication fork barrier in promoting inverted-repeat recombination. Previous studies have shown that the Escherichia coli Tus/Ter complex can function as a DNA replication fork barrier when engineered into the genome of yeast or mouse cells[30–32]. Here, we demonstrate that a polar replication fork barrier engineered to induce fork stalling downstream of inverted repeats is sufficient to trigger NAHR. Physical analysis of the recombinants showed that gene conversion and inversion events are stimulated to the same extent. Unlike spontaneous events, we found that replication-associated NAHR unexpectedly relies on a unique pathway dependent on Rad51 strand invasion and Rad52-Rad59 strand annealing activities. We discuss a model to account for dependence on both Rad51 and Rad52-Rad59 and the formation of gene conversion or inversion outcomes.

## Results

**A polar replication fork barrier stimulates NAHR.** To assess NAHR, we used a recombination reporter composed of two *ade2* heteroalleles oriented as inverted repeats[26]. The inverted-repeat cassette was inserted at the *HIS2* locus, 4 kb centromere distal to the efficient *ARS607* replication origin, on chromosome 6. The origin-proximal *ade2-n* allele contains a +2 frameshift located 370 bp away from the stop codon and is transcribed by the native *ADE2* promoter. The origin-distal allele, *ade2Δ5'*, has a deletion of the first 176 nucleotides along with the promoter. The two repeats share 1.8 kb of homology and are separated by 1.4 kb containing a *TRP1* gene transcribed by its native promoter (Fig. 1a).

To analyze recombination in the context of a unique stressed replication fork, in the absence of any genome-wide stress or global checkpoint activation, we took advantage of the galactose-inducible Tus/Ter replication fork barrier[30,33]. We inserted 14 *TerB* repeats (hereafter referred to as 14 *Ter*) in the permissive or blocking orientation relative to *ARS607*, 120 bp or 170 bp distal to the *ade2Δ5'* repeat, respectively (Fig. 1a). The location was selected based on a previous study showing that Tus/Ter induces mutagenesis of the newly replicated region behind the stalled fork[34]. The $P_{GAL1}$-*Tus* cassette was integrated at the *LEU2* locus.

In cells containing 14 *Ter* repeats in the blocking orientation, an elevated proportion of colonies developing white sectors, indicative of an Ade$^+$ phenotype, was noticeable on plates containing galactose (Fig. 1b). Consistently, quantification of

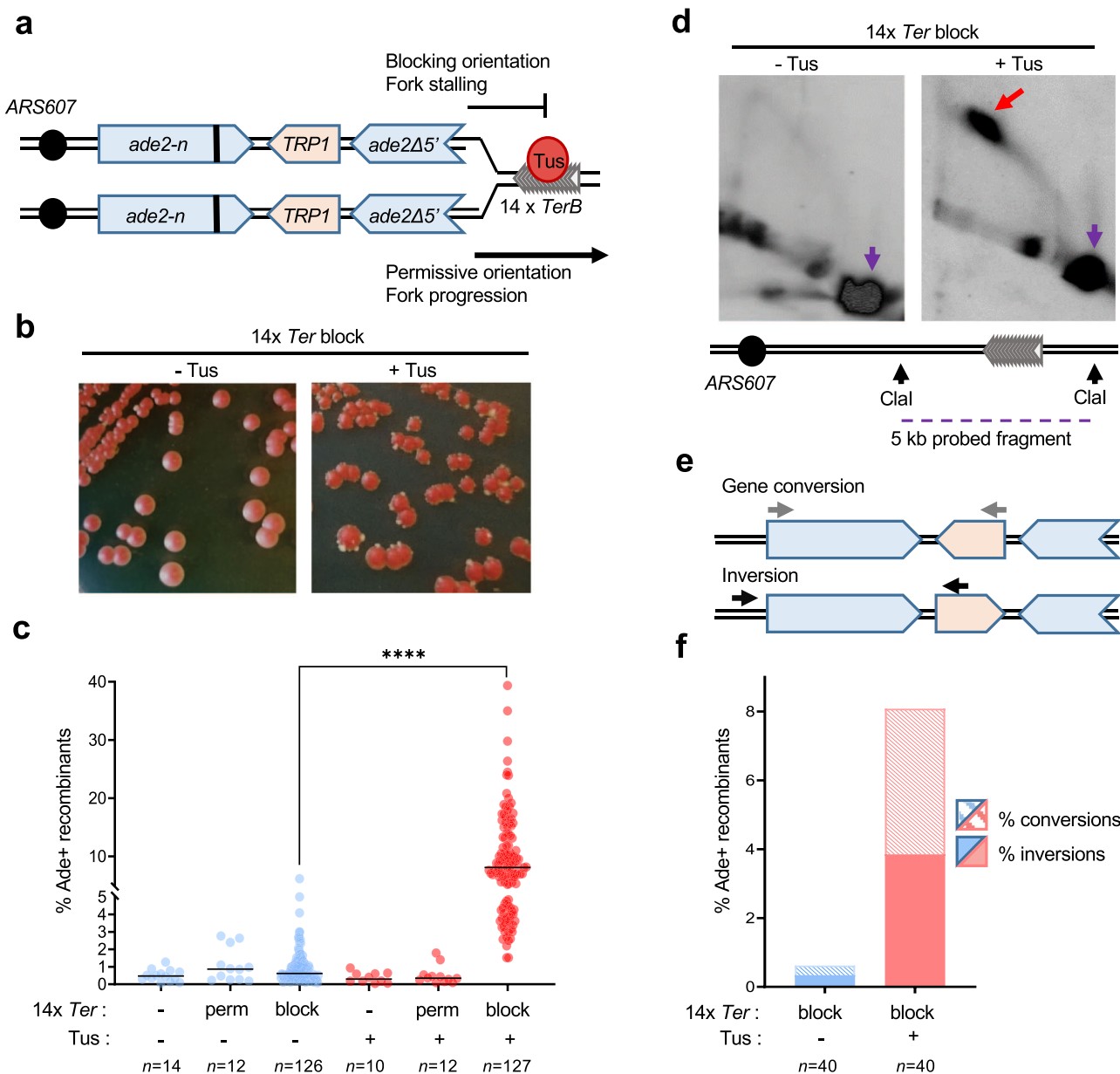

**Fig. 1 A localized fork stalling barrier stimulates NAHR. a** Schematic of the *ade2* reporter and Tus/*Ter* barrier in the blocking or permissive orientation with regards to *ARS607*. The bold line indicates the +2 frameshift mutation. **b** Colonies form more white sectors and papillae (indicative of Ade+ phenotype) when Tus expression is induced. **c** Frequency of Ade+ recombinants without (blue data points) and with (red data points) induction of Tus expression in strains containing different *Ter* constructs (Supplementary Table 1). Black lines indicate medians. *p*-values were obtained on log-transformed data by one-way ANOVA with a Bonferroni post-test and are reported as stars when significant: ****$p$-value < 0.0001. Perm permissive orientation; block blocking orientation. *n* indicates the number of colonies tested. Exact *p*-values are reported in Supplementary Data 1. **d** Two-dimensional gel analysis of replication intermediates in the strain containing 14 *Ter* repeats in the blocking orientation, with or without induction of Tus expression (see Supplementary Fig. S1 for details of probes used). The red arrow indicates fork arrest along the Y-shaped replication arc, the purple arrow indicates the unreplicated fragment. **e** Ade+ recombinants formed by gene conversion or by inversion of the *TRP1* locus are distinguished by PCR using primers designated by gray or black arrows. Inversion events can have the wild type or +2 frameshift site within the *ade2Δ5′* allele and are not distinguished here. **f** Distribution of NAHR events for each condition. *n* indicates the number of independent Ade+ recombinants tested. Data were analyzed by Chi-square test and exact *p*-values are reported in Supplementary Data 1. Source data are provided as a Source Data file.

Ade+ recombinants arising in this strain showed that expression of the Tus protein stimulated recombination frequency from 0.62 to 8.08% (Fig. 1c; Supplementary Table 1). We confirmed that the induction of Tus protein expression had no effect on recombination frequency in cells containing no Ter repeats or 14 *Ter* repeats in the permissive orientation (Fig. 1c and Supplementary Table 1). By two-dimensional (2D) gel analysis of a 5 kb fragment encompassing part of the *ade2* reporter and the *Ter* repeats, we confirmed that induction of Tus protein expression generates a significant replication fork arrest in the strain containing 14 *Ter* repeats in the blocking orientation (Fig. 1d and Supplementary Fig. 1). Thus, replication fork stalling at a polar Tus/*Ter* barrier stimulates recombination between inverted repeats, more than 10-fold. We investigated the nature of the Tus/*Ter*-induced events by a PCR-based method (Fig. 1e). Gene conversions and inversions were equivalently induced upon expression of the

Tus protein, representing 47.5% and 52.5% of the Ade+ recombinants, respectively (Fig. 1f).

The role of genome-wide replication stress in the stimulation of NAHR was assessed by growing cells with the *ade2* reporter on media containing DNA damaging agents known to induce replication stress, namely, methyl methanesulfanate (MMS), camptothecin (CPT), or hydroxyurea (HU). Within three days, an increased proportion of colonies containing white sectors, indicative of an Ade+ phenotype, was clearly visible in the presence of MMS and CPT (Supplementary Fig. 2a). Consistently, quantification of Ade+ recombination frequencies under normal conditions (0.62% spontaneous recombination) and genotoxic conditions (16.15% with MMS, 9.79% with CPT, 1.5% with HU) revealed a strong stimulation of recombination between the inverted *ade2* repeats in presence of MMS and CPT (Supplementary Fig. 2b). The types of recombination events induced by MMS or CPT were determined by PCR analysis of independent recombinants. In the presence of MMS, the frequency of gene conversions was 38-fold higher (10.8%), whereas the frequency of inversions was increased by a factor 16 (5.4%). In presence of CPT, the frequency of gene conversions was 11 times higher (3.2%), whereas inversions were induced 19-fold (6.36%) (Supplementary Fig. 2c). We note that in the presence of CPT, the nature of the recombination event of some recombinants could not be easily determined by the PCR method employed (2% of tested recombinants) and these were not analyzed further. We detected a moderate induction of recombination frequency by HU (2.4-fold induction) and the distribution of gene conversions and inversions appeared similar to normal conditions (Supplementary Fig. 2b, c). A 3 h liquid incubation with high HU concentrations to completely halt replication did not result in a stronger induction of recombination (Supplementary Fig. 2d).

Together, these results indicate that NAHR between long inverted repeats, leading to gene conversion or inversion of the intervening sequence, can be generated by genome-wide replication stress or by a localized replication fork barrier, consistent with prior studies in *S. pombe* and mouse cells[22,24,32].

**Replication-associated NAHR has unique genetic requirements.** In line with previous studies[27,29], we found that spontaneous gene conversions and inversions are products of two independent recombination pathways. In the absence of Tus/*Ter*-induced replication stress, deletion of *RAD51* or *RAD59* only partially decreased recombination between the *ade2* repeats, whereas no recombination was detected in the double mutant. The Rad52 protein is involved in both pathways as the recombination frequency of the *rad52Δ* strain, like the *rad51Δ rad59Δ* double mutant, was below detection (Fig. 2a and Supplementary Table 1). Physical analysis of spontaneous Ade+ recombinants arising in the *rad51Δ* mutant showed that 84% of the tested recombinants contained an inversion. On the other hand, in the *rad59Δ* mutant, 67% of the recombinants were gene conversions (Fig. 2b). These results confirm that spontaneous NAHR events leading to inversions of the *TRP1* gene are largely independent of Rad51 and require Rad59 and Rad52, whereas gene conversions are mostly independent of Rad59 and require Rad51 and Rad52[27].

Surprisingly, both *rad51Δ* and *rad59Δ* single mutants showed no induction of recombination by the Tus/*Ter* replication barrier (Fig. 2c). We confirmed by 2D gels that the Tus-generated replication fork barrier was still detected in both mutants (Fig. 2d). Thus, it appears that gene conversions and inversions associated with replication fork stalling have specific genetic requirements. Rad52 is essential for recombination in this context as well since no Ade+ recombinants were detected in the *rad52Δ*

strain (Fig. 2c). Intriguingly, physical analysis of Tus-induced recombinants in the *rad51Δ* and *rad59Δ* mutant strains revealed a different distribution from spontaneous events (Fig. 2e).

We also determined the frequency of MMS and CPT-stimulated recombination in *rad51Δ* and *rad59Δ* single mutants using low concentrations of the drugs that allowed the growth of the mutants while stimulating recombination in the WT strain (Supplementary Fig. 2e, f). Whereas the frequency of recombination in the WT strain increased from 0.62 to 14.38% with MMS and 2.09% with CPT, we detected no stimulation of recombination by genotoxic agents in *rad51Δ* and *rad59Δ* mutants.

Together, these results indicate that recombination between inverted repeats associated with replication stress is mediated by a unique molecular mechanism involving Rad51, Rad52, and Rad59, and leads to both gene conversions and inversions (Fig. 2f).

**Replication-associated NAHR relies on strand invasion and annealing.** We next wanted to further explore the roles of Rad51 and Rad52 in Tus/*Ter*-induced NAHR. Rad51 has three established functions at stalled replication forks. First, Rad51 promotes replication fork reversal in mammalian cells, but does not have fork remodeling activity on its own and different models have been proposed to explain its role in this process[35]. Second, Rad51 is required for the protection of nascent DNA strands at reversed forks from extensive nucleolytic degradation by Mre11[36,37]. Finally, Rad51 plays a role in the restart of arrested replication forks by several recombination pathways involving strand invasion and strand exchange[35,37,38].

The *rad51-II3A* allele contains three amino acid substitutions, eliminating the secondary DNA binding site. The mutant protein retains the ability to form filaments on ssDNA but is defective for strand exchange activity[39,40]. A recent study, modeling this mutation in human cells, revealed that the enzymatic activity of Rad51 is neither required to promote fork reversal nor to protect stalled forks from extensive degradation. In contrast, efficient replication restart is dependent on Rad51 strand exchange activity, but can be partially rescued by strand exchange-independent mechanisms such as regression of the reversed fork by branch migration or replication origin firing[40]. Similarly, the *rad51-II3A* mutant protects stalled replication forks from nucleolytic degradation in *S. pombe*[41].

We introduced the *rad51-II3A* allele in the strain containing the *ade2* inverted repeats and 14 *Ter* repeats in the blocking orientation. In the absence of Tus, the frequency of spontaneous recombination decreased from 0.62% in the WT to 0.21% in the mutant (Fig. 3a, blue data points). Furthermore, induction of fork stalling did not stimulate recombination between the *ade2* inverted repeats (Fig. 3a, red data points). We note that although not statistically significant (*p*-value = 0.3), spontaneous and replication-associated recombination in the *rad51Δ* strain was a little higher than in the *rad51-II3A* mutant which might indicate that the presence of inactive rad51-II3A filaments limits recombination events.

Rad52 has two functions in homologous recombination: mediation of Rad51 nucleoprotein filament assembly on RPA-coated ssDNA and annealing of complementary ssDNA during second end capture or SSA[25]. The rad52-R70A separation-of-function mutant is proficient for Rad51 loading but defective for ssDNA annealing[42]. We observed a 22-fold decreased frequency of Tus-induced recombination in the *rad52-R70A* mutant strain, consistent with an important role for strand annealing during replication-associated NAHR (Fig. 3b).

Taken together, these results suggest that HR associated with fork stalling relies on Rad51-catalyzed strand invasion, distinct

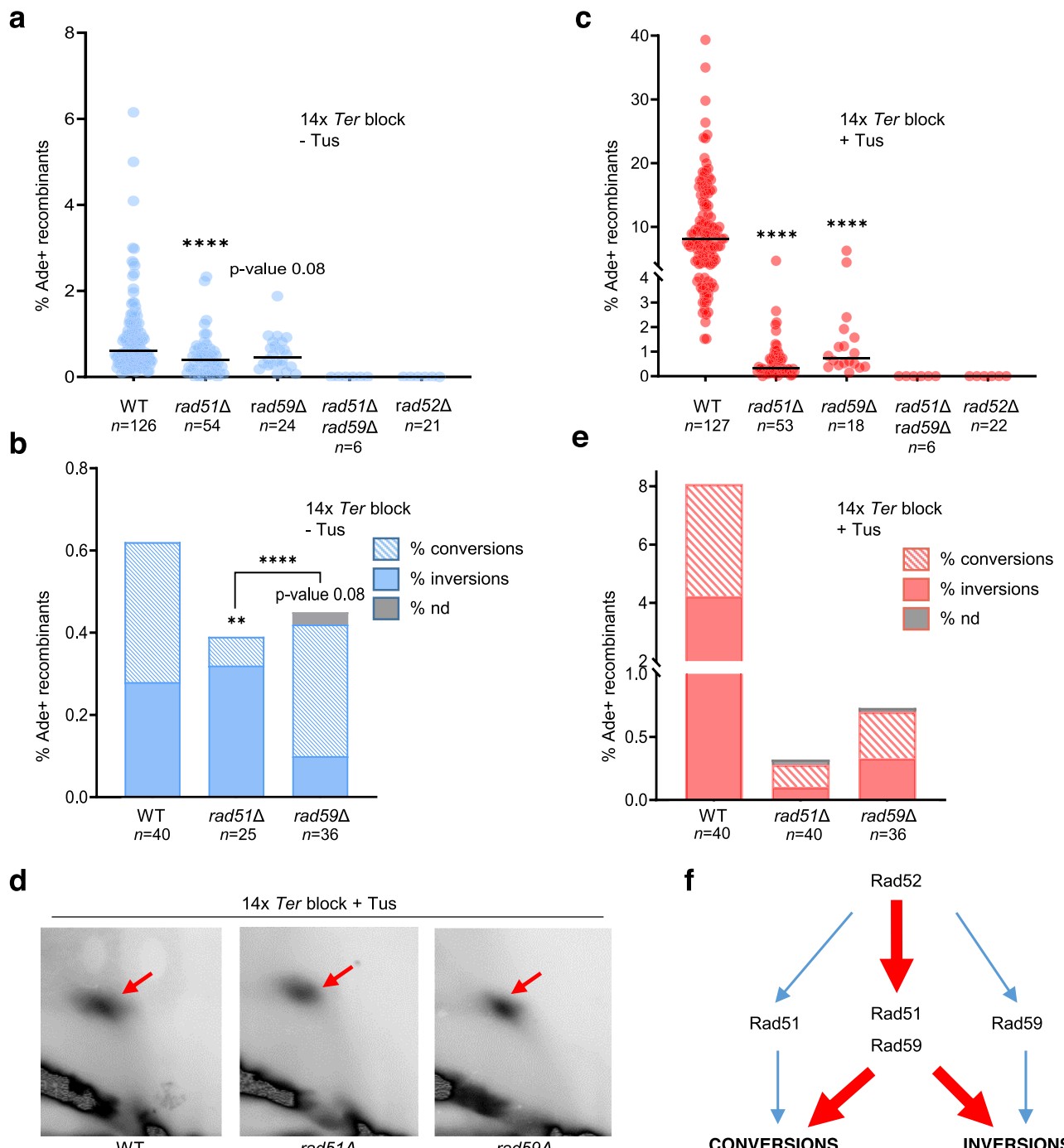

**Fig. 2 NAHR at the Tus/Ter barrier relies on the cooperation of Rad51 and Rad59. a** Spontaneous Ade+ recombination frequencies in WT and mutant strains. Black lines indicate medians; *n* indicates the number of colonies tested. *p*-values were obtained on log-transformed data by one-way ANOVA with a Bonferroni post-test and are relative to the WT strain in the same condition. They are reported as stars when significant with: ****p*-value < 0.0001. Exact *p*-values are reported in Supplementary Data 1. **b** Distribution of independent spontaneous recombination events scored by PCR; nd indicates structure could not be determined by PCR; *n* indicates the number of independent Ade+ recombinants tested. *p*-values, reported as stars when significant, were obtained by Chi-square test: ***p*-value < 0.005, ****p*-value < 0.0001. Exact *p*-values are reported in Supplementary Data 1. **c** Tus-induced Ade+ recombination frequencies in WT and mutant strains. Black lines indicate medians; *n* indicates the number of colonies tested. *P*-values were obtained on log-transformed data by one-way ANOVA with a Bonferroni post-test and are relative to the WT strain in the same condition. They are reported as stars when significant with ****p*-value < 0.0001. **d** 2D gel analysis of replication intermediates showing similar fork arrest in the WT and mutant strains. Red arrows indicate the replication fork arrest on the arc of Y-shaped replication intermediates. **e** Distribution of events scored by PCR for Tus-induced recombinants. *n* indicates the number of independent Ade+ recombinants tested. Data were analyzed by Chi-square test. Source data are provided as a Source Data file. **f** Contribution of Rad51, Rad52, and Rad59 to spontaneous (blue) and replication fork block-induced (red) NAHR pathways.

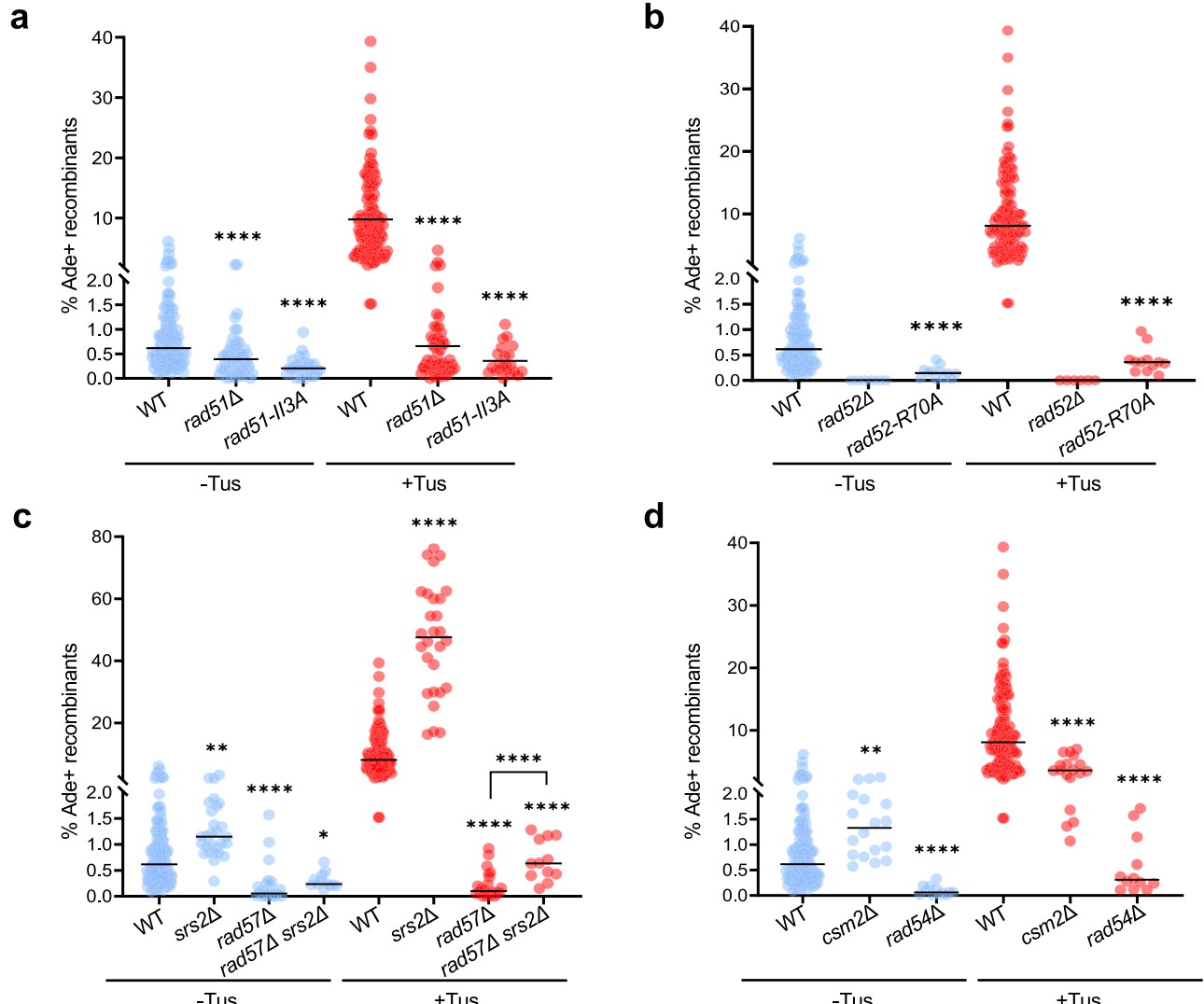

**Fig. 3 Rad51 strand invasion, Rad52 strand annealing, and recombination mediators are required for Tus/Ter-induced NAHR. a** Ade+ recombination frequencies for WT and rad51 mutant strains containing 14 Ter repeats in the blocking orientation. **b** Ade+ recombination frequencies in WT and rad52 mutant strains containing 14 Ter repeats in the blocking orientation. **c** Ade+ recombination frequencies in WT, srs2Δ and rad57Δ single and double mutant strains containing 14 Ter repeats in the blocking orientation. **d** Ade+ recombinants frequencies in WT, cms2Δ and rad54Δ mutant strains containing 14 Ter repeats in the blocking orientation. **a–d** Black lines indicate medians. The numbers of colonies tested for each strain are reported in Supplementary Table 1. p-values were obtained on log-transformed data by one-way ANOVA with a Bonferroni post-test and are relative to the WT strain in the same condition. They are reported as stars when significant with: *p-value < 0.05, **p-value < 0.005, ***p-value < 0.001, ****p-value < 0.0001. Source data are provided as a Source Data file and exact p-values are reported in Supplementary Data 1.

from its role in protecting stalled forks from degradation, as well as Rad52-Rad59 catalyzed strand annealing.

**Spontaneous and replication-associated NAHR involve distinct mediators.** We next assessed the contribution of various Rad51 mediators in spontaneous and Tus/Ter-induced recombination. The Rad51 paralogs, Rad55 and Rad57, form a stable heterodimer that assists Rad51 nucleation on RPA-coated ssDNA and promotes rapid re-assembly of filaments after their disruption by the Srs2 anti-recombinase[43,44]. Spontaneous recombination between the ade2 repeats was reduced by 11-fold in the rad57Δ mutant strain, as opposed to only 1.6-fold in the rad51Δ strain, consistent with a previous study (Fig. 3c, blue data points)[45]. This finding could indicate that in the absence of the Rad55-Rad57 complex to stabilize Rad51, unstable Rad51 filaments are unable to mediate gene conversion but also inhibit the Rad51-independent

spontaneous inversion pathway. When replication fork stalling was induced, there was no stimulation of recombination in the rad57Δ strain (Fig. 3c, red data points) and recombination was again more deficient in the rad57Δ strain than it was in the rad51Δ mutant (0.09% vs 0.32%). We also tested whether loss of Srs2 suppresses the rad57Δ defect in Tus/Ter-induced recombination. Consistent with previous studies[46], spontaneous recombination was increased in the srs2Δ mutant, and Tus/Ter-stimulated recombination was increased by 5-fold over the WT value (Fig. 3c). Loss of Srs2 partially rescued the rad57Δ recombination defect, but the frequency was still 10-fold lower than WT cells, indicating that Rad57's function is not restricted to antagonizing Srs2.

The Shu complex is another mediator of Rad51 presynaptic filament formation, which interacts directly with Rad51 and Rad55-Rad57, and has been specifically implicated in the repair of DNA replication-associated damage[47–49]. Csm2 is one of the four

members of the Shu complex. Unlike in the *rad57Δ* mutant, spontaneous recombination between the repeats was not diminished and was even moderately enhanced in the *csm2Δ* mutant (Fig. 3d, blue data points). However, when replication fork stalling was induced, recombination in the *csm2Δ* mutant was two-fold lower than in the WT strain (Fig. 3d, red data points) suggesting that the Shu complex facilitates NAHR at stalled replication forks but is not strictly required.

Rad54 is an ATP-dependent dsDNA translocase that is required to facilitate Rad51-mediated strand invasion[50]. Consistent with a previous study[45], we observed that spontaneous recombination was significantly reduced in the *rad54Δ* mutant, and Tus-induced events were 26-fold lower than WT (0.31% vs 8.08%), similar to the frequency observed for the *rad51Δ* mutant (Fig. 3d).

Our results show that Rad51 and its mediators are differentially implicated in spontaneous and replication-associated inverted-repeat recombination. These data indicate that replication-associated NAHR must involve invasion of ssDNA from one *ade2* copy into dsDNA from the other *ade2* copy. Only the long *ade2-n* allele can be restored to a functional *ADE2* gene, so we reasoned that the truncated copy must be the one invaded and used as a donor template. Based on the position of the replication fork barrier in our genetic system, fork reversal would promote reannealing of the parental strands of the truncated *ade2* copy, thus providing a dsDNA substrate for invasion.

**Is fork reversal required for replication-associated NAHR?** To determine the role of fork reversal in Tus/*Ter*-stimulated recombination we eliminated DNA remodelers that have been implicated in fork reversal. The translocase Rad5 (HLTF in human) initiates replication fork reversal by remodeling the leading strand and proximally positioning the leading and lagging arms, which converts the arrested fork into a chicken-foot structure[51–55]. However, deletion of *RAD5* showed no significant effect on spontaneous or replication-associated recombination (Fig. 4a). The Mph1 helicase (FANCM in human, Fml1 in *S. pombe*) also promotes fork reversal in vitro and is required for recombination at a protein-induced fork barrier in *S. pombe*[56,57]. Loss of Mph1 did not reduce the frequency of spontaneous recombination; however, Tus/*Ter*-stimulated recombination was moderately reduced (*p*-value = 0.06), and recombination was further reduced in the *mph1Δ rad5Δ* double mutant (Fig. 4a). Taken together, our results suggest that NAHR events associated with fork stalling require remodeling activity of Mph1 with Rad5 serving a minor or redundant function. Physical analysis of independent recombinants in the *mph1Δ* mutant showed a distribution of replication-associated events similar to the WT strain (Supplementary Fig. 3a). However, in the *mph1Δ rad5Δ* double mutant conversions were reduced 15-fold compared to the WT strain, whereas inversions were only reduced 4-fold (Supplementary Fig. 3a). This could be due to an additional effect of Mph1, in this context, in dissociating the migrating D-loop, thus leading to proportionally more inversions in the *mph1Δ rad5Δ* double mutant. Alternatively, ssDNA gaps formed at the stalled fork in the absence of Mph1-catalyzed fork reversal could be acted on by Rad5-dependent template switching to produce gene conversion recombinants.

**Tus/Ter-stimulated recombination requires resection machineries.** Fork reversal at the Tus/*Ter* stall is predicted to form a single-ended DSB, which could be converted to a ssDNA substrate for Rad51 loading by end resection. DNA end resection occurs by a two-step mechanism involving sequential action by short-range and long-range resection nucleases[58]. Mre11 nuclease

initiates end resection at DSBs as part of the Mre11-Rad50-Xrs2 complex, while Exo1 or Dna2-Sgs1 promotes extensive resection. Studies in *S. pombe* and in mammalian cells have shown that the same nucleases can degrade regressed forks[2,59]. In budding yeast, MRX is essential for resection of DSBs with end-blocking lesions, but resection can still occur at "clean" DSBs by the direct action of the long-range nucleases, Exo1 and Dna2[58].

To determine the role of DNA end resection in Tus/*Ter*-stimulated recombination, we eliminated DNA nucleases that function in short and long-range resection. The frequencies of spontaneous and Tus-induced recombination were reduced by 3-fold in the *mre11Δ* mutant (Fig. 4b) indicating a role for resection initiation by MRX. In the absence of Exo1, spontaneous recombination occurred at the WT level (Fig. 4b, blue data points); however, stimulation of recombination by the Tus/*Ter* barrier was abolished (Fig. 4b, red data points). This finding suggests that replication-associated NAHR relies on extensive degradation of the newly synthesized lagging strand by Exo1 to generate a ssDNA leading strand substrate for Rad51 loading.

Sgs1 and Dna2 act redundantly with Exo1 in long-range resection at DSBs. We found that the frequency of Tus/*Ter*-stimulated recombination was not reduced in the *sgs1Δ* mutant compared to the WT (Fig. 4b, red data points). In the absence or presence of Tus expression, *sgs1Δ* cells showed a slight increase in recombination (Fig. 4b, blue data points), consistent with the previously reported hyper-recombination phenotype[60]. This result seems to indicate that Sgs1 does not play a role in fork resection. However, the caveat is that Sgs1 is involved in other processes, such as the dissolution of recombination intermediates, and these roles could mask a role in fork resection[61]. To address this possibility, we investigated the role of *DNA2* in NAHR events. In *S. cerevisiae*, deletion of *DNA2* is lethal but can be rescued by the *pif1-m2* allele[62]. We introduced *pif1-m2* or *dna2Δ pif1-m2* alleles into the *ade2* inverted-repeat reporter strain. In the absence of Tus, we did not detect any differences in the frequency of spontaneous recombination between the WT and the mutant strains (Fig. 4b, blue data points). When Tus was expressed, the *pif1-m2* strain showed an induction of recombination comparable to the WT strain. However, we observed a 3.4-fold decreased frequency of Tus-induced recombination in the *dna2Δ pif1-m2* double mutant as compared to the *pif1-m2* single mutant (Fig. 4b, red data points). These data are consistent with Sgs1-Dna2-dependent long-range resection contributing to replication-associated NAHR.

**Opposing roles of structure-selective nucleases.** Fork reversal at the Tus-induced barrier could generate an invading end with a short sequence heterology that would need to be removed to prime DNA synthesis within the D-loop intermediate. Previous studies in yeast have shown that Rad1-Rad10 nuclease removes 3′ heterologies during Rad51-dependent strand invasion, as well as 3′ flaps formed during Rad51-independent SSA[25]. Consistent with the need for heterologous flap or loop removal, the frequency of Tus/*Ter*-induced recombination was reduced by 8-fold in the *rad1Δ* mutant (Fig. 4c, red data points).

Fork reversal creates a four-way junction that can be cleaved by structure-selective nucleases to create a one-ended DSB. In budding yeast, Mus81-Mms4 is the main nuclease responsible for cleaving recombination intermediates, with Yen1 providing a back-up function[63–65]. We did not find a significant change in the frequency of spontaneous or replication-associated recombination in the *mus81Δ* mutant. However, elimination of Yen1 and Mus81 resulted in a 2-fold increase in the frequency of Tus/*Ter*-induced recombination from 8.08% to 15.45% (Fig. 4c). Thus, Mus81-Mms4 and Yen1 may abort the normal process for

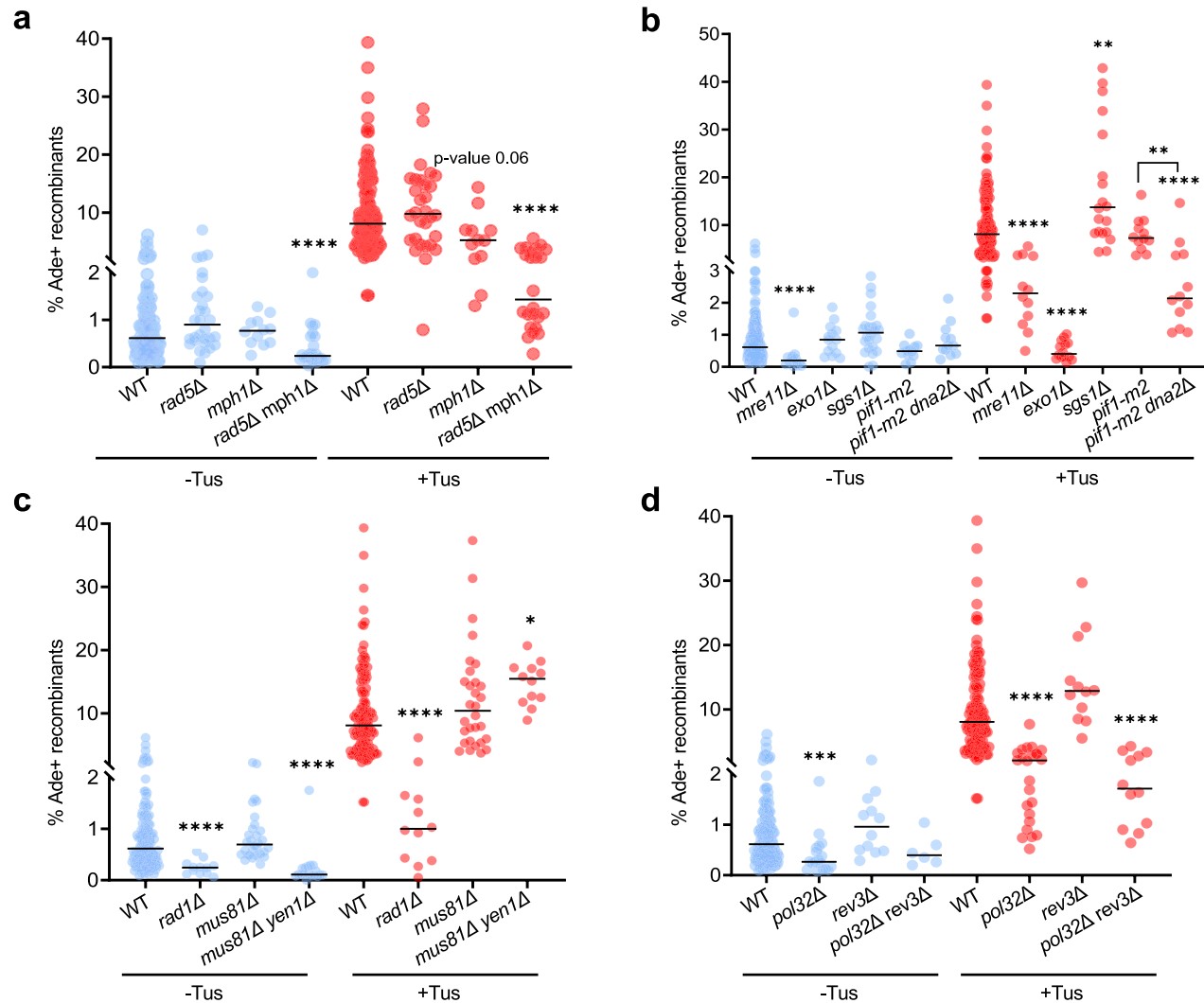

**Fig. 4 Tus-induced NAHR relies on fork reversal, end resection, and DNA synthesis. a** Frequency of Ade+ recombinants in WT and fork remodeler mutant strains. **b** Ade+ recombination frequencies in WT and end resection mutants. **c** Ade+ recombination frequencies in WT and nuclease mutants. **d** Ade+ recombination frequencies in WT and polymerase mutants. **a–d** Black lines indicate medians. The numbers of colonies tested for each strain are reported in Supplementary Table 1. *p*-values were obtained on log-transformed data by one-way ANOVA with a Bonferroni post-test and are relative to the WT strain in the same condition. They are reported as stars when significant with: \**p*-value < 0.05, \*\**p*-value < 0.005, \*\*\**p*-value < 0.001, \*\*\*\**p*-value < 0.0001. Source data are provided as a Source Data file and exact *p*-values are reported in Supplementary Data 1.

forming recombinants at the Tus/Ter barrier. We also looked at the distribution of replication-associated recombination events in the *mus81Δ yen1Δ* double mutant (Supplementary Fig. 3b). Inversions represent more than 50% of the products in the double mutant indicating that they are not generated by cleavage of a Holliday junction (HJ)-containing intermediate. The increase in Tus/Ter-stimulated inversion products in the *mus81Δ yen1Δ* double mutant suggests that Mus81-Mms4 and Yen1 might cleave the migrating D-loop initiated by Rad51, in addition to the reversed fork intermediate.

**A specific role for polymerase Pol δ in replication-associated NAHR.** NAHR is predicted to require DNA synthesis to convert the *ade2-n* allele, and potentially to invert the *TRP1* gene between the repeats. In vivo and in vitro studies have shown that DNA Pol δ initiates synthesis from the invading 3′ end within the D-loop intermediate[66–68]. *S. cerevisiae* Pol δ is a heterotrimer comprised of a catalytic subunit Pol3 and two accessory subunits Pol31 and

Pol32. Pol31 and Pol32 also associate with Rev3 and Rev7 to form another B-family DNA polymerase, Pol ζ, a translesion polymerase responsible for mutagenic replication of damaged DNA[69].

When we deleted *POL32* in the *ade2* reporter strain containing 14 *Ter* repeats in the blocking orientation, we observed a decrease in the frequency of spontaneous recombination (Fig. 4d, blue data points). Upon induction of fork stalling by Tus/Ter in the *pol32Δ* mutant, we observed a significant decrease of recombination compared to the WT strain (2.08% vs 8.08% in WT) (Fig. 4d, red data points). Physical analysis of Tus/Ter-induced recombinants showed a similar distribution to the WT strain (Supplementary Fig. 3c). To determine whether the *pol32Δ* defect was due to Pol δ or Pol ζ, we measured recombination frequencies in a *rev3Δ* mutant (Fig. 4d). Unlike the *pol32Δ* strain, the *rev3Δ* mutant showed a full stimulation of recombination upon induction of the Tus/Ter barrier. The double mutant exhibited a similar phenotype to the *pol32Δ* single mutant; thus, Pol δ but not Pol ζ appears to be involved in this process.

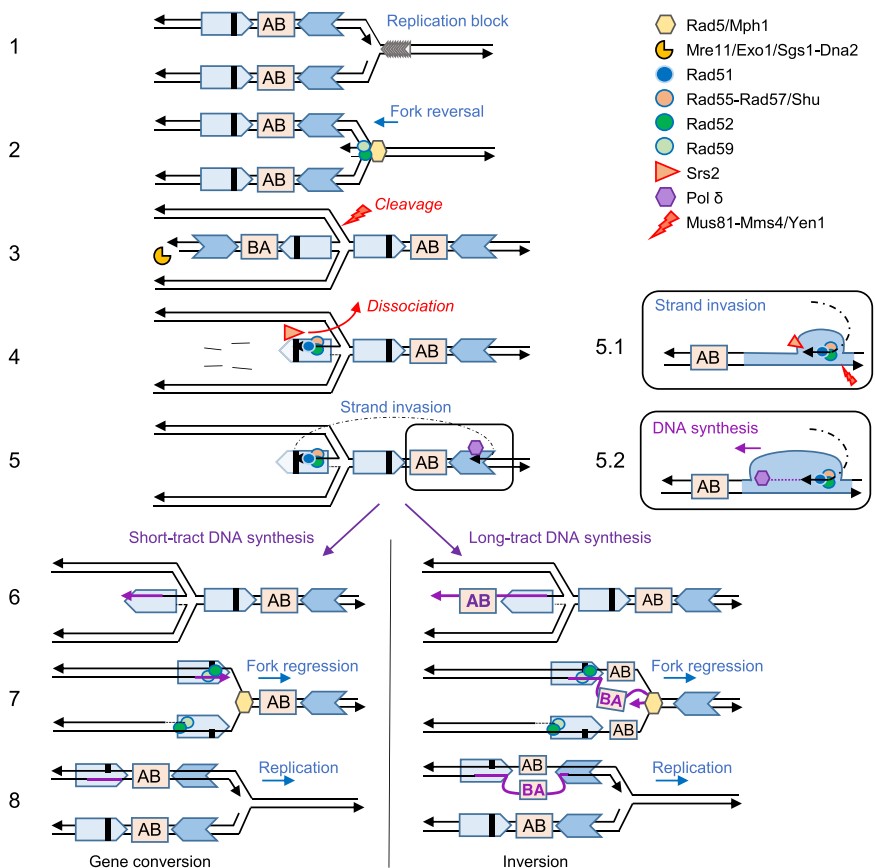

**Fig. 5 Model for NAHR at arrested replication forks.** (1) The replication fork stalls at the Tus/*Ter* barrier. The letters AB indicate the orientation of the intervening sequence. The bold black line represents the +2 frameshift. (2) Rad5 and Mph1 catalyze fork reversal. Rad52-Rad59 annealing activity could facilitate strand pairing of the daughter or parental strands. (3) The regressed arm is degraded by short and long-range resection machineries. The reversed fork can be cleaved by nucleases, aborting NAHR. (4) Rad51 polymerization on ssDNA is mediated by Rad52 and Rad55-Rad57 (with help from the Shu complex), counteracting the Srs2 anti-recombinase. (5) Rad51 catalyzes strand invasion into the parental non-allelic repeat, heterologies are cleaved by Rad1-Rad10 and DNA synthesis is initiated by Pol δ. The D-loop can be dissociated by Mph1 or Srs2, or cleaved by Mus81-Mms4 and Yen1. (5.1) Zoom on the structures proposed to form during strand invasion. (5.2) Zoom on DNA synthesis initiation. (6) Short-tract DNA synthesis leads to gene conversion, whereas long-tract DNA synthesis leads to the inversion of the intervening sequence (AB to BA) on the newly synthesized leading strand. (7) Regression of the reversed fork. (8) Replication restarts. Heteroduplex DNA incorporating the frameshift mutation could be corrected by mismatch repair to produce a gene conversion product or segregated at the next S phase. The large unpaired heterologous loop could be repaired by Rad1-Rad10 or segregated at the next replication cycle.

## Discussion

Replication stress, defined as a slowing down or complete arrest of DNA synthesis during chromosome replication, has emerged as a primary cause of genomic instability, a hallmark of cancer and other human disorders associated with genomic rearrangements[1,70,71]. In fission yeast and mammalian cells, replication fork stalling adjacent to a recombination reporter can lead to increased recombination events[32,72]. In this work, we show that a Tus/*Ter* barrier designed to induce transient replication fork stalling near inverted repeats stimulates recombination mediated by a unique genetic pathway, distinct from spontaneous NAHR or post-replicative repair. The model presented in Fig. 5, which is discussed in detail below, is based on our genetic findings and builds on other template switching models for replication-associated recombination.

Spontaneous inversion of the sequence between inverted repeats is dependent on Rad52 and Rad59, whereas spontaneous gene conversion without an associated inversion is dependent on Rad52 and Rad51[26,29]. In contrast, we show here that HR induced at a replication fork barrier triggers both inversions and gene conversions mediated by Rad51, Rad59, and Rad52 working together in a unique pathway. The template switching mechanism of post-replicative

repair (PRR) is a DNA damage tolerance pathway that involves use of the undamaged sister chromatid as a homologous template for lesion bypass[4]. One model of template switching involves the reversal of the stalled fork for stabilization and/or repositioning of the lesion to bypass damage[5,73]. The other model involves the pairing of a template strand at a ssDNA gap with the undamaged sister chromatid to form a pseudo-double HJ intermediate[4]. The second mode of PRR template switching is mediated by several proteins that we show are also important for NAHR at Tus/*Ter* stalled forks: Rad51, Rad55-Rad57, Csm2, Exo1, and DNA Pol δ[74,75]. However, Rad59 is not required for PRR template switching, whereas we detected a strong reduction of NAHR at stalled forks in the *rad59Δ* mutant[74]. In addition, Rad5, which is essential for PRR template switching[75], is only required for Tus-induced recombination in the absence of Mph1. Overall, our data show that the mode of HR associated with replication stalling at repetitive sequences is genetically different from spontaneous HR or PRR template switching pathways.

Rad59 contributes to a subset of HR events by assisting Rad52 in second end capture during DSB repair and in SSA at direct repeats[76,77]. Thus, Rad59-dependent recombination is thought to be linked to DSB repair where both ends have to be rescued through simultaneous interactions with an unbroken template.

However, reversal of the stalled fork at the Tus/Ter barrier would generate a regressed arm resembling a one-ended break with no second end for capture by Rad52-Rad59 mediated strand annealing. Thus, the sizable decrease in Rad51-mediated HR at Tus/Ter in the rad59Δ mutant is intriguing. We suggest that Rad59 acts with Rad52 in facilitating regression of the stalled fork, or restoration of the reversed fork, by mediating annealing of nascent or parental strands. Interestingly, the role of mammalian RAD52 has long remained mysterious due to the presence of BRCA2, which assumes RAD51-mediator function and prevents any significant DNA repair phenotype in RAD52-deficient cells. However, RAD52 was recently shown to have a specific protective role in maintaining cell viability under replication stress that is non-redundant with BRCA2[78,79]. Our results suggest a conserved function for Rad52 during replication stress, involving its strand annealing activity.

Rad59 could also function by stabilizing an annealed intermediate with a heterologous tail for cleavage by Rad1-Rad10, as previously suggested[80]. Such an intermediate could occur after fork resetting (Fig. 5). The other possible functions of Rad1-Rad10 could be in the repair of the large loop heterology expected to occur from long-tract synthesis and fork reset[81].

We did not detect a decrease of replication-associated NAHR in the rad5Δ single mutant, again highlighting the specific genetic requirements of this pathway compared to PRR template switching. However, we found a 5.5-fold decrease in Tus-induced recombination in the rad5Δ mph1Δ strain compared to the WT. The relationship between Rad5 and Mph1, the two major DNA remodelers in budding yeast with reported replication fork regression activity, is not fully understood. The additive defect in MMS resistance observed for the double mutant[82], and partial suppression of MMS sensitivity of a rad5Δ mutant by Mph1 hyperactivation, suggests that they have overlapping activities[53], consistent with our findings. The requirement for Rad51 strand invasion activity leads us to propose a model involving invasion of the parental duplex, which would require fork regression to create an invading end. We note that the fork would need to reverse by several kb for the ade2-n allele to be placed for invasion of the reformed parental ade2-5'Δ allele (Fig. 5), which could lead to an under-estimation of recombination induced by the Tus/Ter block. Mph1 is also involved in D-loop dissociation during DSB repair and HR-mediated restart of collapsed replication forks[83–85]. If the main activity of Mph1 during NAHR at stalled forks is to dissociate the D-loop we would not expect to observe a reduction in recombination frequency, although the change in the proportion of inversions in the rad5Δ mph1Δ double mutant is consistent with D-loop dissociation activity of Mph1. It was recently proposed that Mph1 can act coordinately with Rad54 and Rad5 in the HR-driven fork regression mechanism to bypass stalled replication forks[86]. We observed a strong reduction of spontaneous and replication-associated NAHR events in the rad54Δ mutant. This outcome could be due to a role in fork reversal in addition to the role of Rad54 in promoting Rad51-mediated strand invasion[87]. However, based on the phenotype of rad5Δ mph1Δ double mutant, Rad54 does not appear to play a major role in fork reversal.

We envision that a stalled replication fork is reversed into a chicken-foot structure by the redundant activities of Rad5 and Mph1 (and potentially Rad54). We propose that the process is assisted by Rad52 and Rad59 which facilitate nascent strand pairing. Fork reversal creates a branched structure that could be acted upon by endonucleases such as Mus81-Mms4 and Yen1 or counteracted by helicases. The reversed fork exposes a regressed arm which is processed to form a 3′ ssDNA overhang by the sequential activities of Mre11 and Exo1 or Sgs1-Dna2, generating a ssDNA template for Rad51 nucleoprotein filament formation on the leading strand. The long 3′ overhang could be removed by nucleases (as shown in Fig. 5) or be cleaved by Rad1-Rad10 after

the invasion of the other copy. We show that Rad51 loading on the leading ssDNA template is facilitated by Rad55-Rad57 and the Shu complex. One activity of Rad55-Rad57 is to counteract the anti-recombinase Srs2, but our finding that deletion of SRS2 only partially suppresses the rad57Δ HR defect suggests an additional function for the Rad51 paralog complex. Interestingly, a recent study showed that Rad55-Rad57 is essential for the promotion of UV-induced HR independently of Srs2 and prevents the recruitment of translesion synthesis polymerases which would compete with template switching[88].

We propose that Rad51 catalyzes strand invasion into the parental non-allelic inverted sequence ahead of the reversed fork, facilitated by the dsDNA translocase Rad54. DNA synthesis is mediated by Pol δ using the repetitive sequence as a template. Dissociation of the extended invading strand prior to the intervening sequence (represented by AB in Fig. 5) would result in no inversion. Such dissociation may be promoted by Mph1. On the other hand, long-tract DNA synthesis through the intervening sequence (B before A in Fig. 5) could result in its inversion. Theoretically, inversions could also result from cleavage of a HJ intermediate by Mus81-Mms4 or Yen1. However, we did not observe any decrease of inversions in the mus81Δ yen1Δ double mutant (Supplementary Fig. 3b).

The reversed fork would then need to be regressed by the action of remodelers and/or strand annealing proteins to restore the replication fork. Regression of the reversed fork could dissociate the D-loop, or helicases could dismantle the D-loop prior to regression. The resulting replication fork would contain heteroduplex DNA encompassing the ade2-n allele with the potential to create a functional ADE2 gene by mismatch correction or segregation of the strands at the next cell cycle. A heterologous tail or loop, formed between the inverted repeats could be cleaved by Rad1-Rad10, or segregate at the next replication cycle resulting in two daughter cells with either a conversion or an inversion.

A prediction from this model is that POL32 deletion, which is expected to decrease the length of the DNA synthesis tracts[89], would result in an increased proportion of conversions. However, analysis of pol32Δ recombinants showed that the distribution of inversions and gene conversions was similar to WT (Supplementary Fig. 3c). We propose an alternative model for the late stages of replication-associated NAHR after fork resetting (Supplementary Fig. 4). Heterologous tails, which might be formed by short-tract DNA synthesis that extends beyond the region of shared homology, could provide an entry point for helicases and nucleases leading to the degradation of the newly synthesized strand effectively aborting recombination. Most recombinants arising in our assay would result from long-tract DNA synthesis generating a heterologous loop between the repetitive sequences on fork resetting. The four-fold decrease in the frequency of recombinants in the pol32Δ strain could reflect the minority of DNA synthesis tracts long enough to span the sequences between the ade2 repeats[89]. The equal proportions of inversion and gene conversion events in the WT strain are consistent with the repair mechanism depicted in Supplementary Fig. 4b, right panel. We note that, although this alternate model would explain why the two different recombination outcomes are similarly reduced in the pol32Δ mutant strain, degradation of the heterologous tail due to short-tract synthesis would have to extend several hundreds of bp into the full-length repeat to reach the frameshift position.

In conclusion, this work uncovers a genetically unique pathway that is stimulated by localized replication stress and can mediate genomic rearrangements of repetitive sequences. It should be noted that our reporter system can only reveal recombination events that lead to the restoration of a functional ADE2 gene. If strand invasion of the non-allelic copy occurred downstream from the +2 frameshift location a recombinant would not be detected. The frequency of

NAHR events at inverted repeats that can generate rearrangements is thus likely to be underestimated in our assay.

How spontaneous Rad51-independent inversions are generated remains an open question to be explored. Previous studies suggest that they are not due to DSB repair and this work supports the idea they are not associated with fork stalling at a protein barrier. Other contexts that could be investigated are replication uncoupling to form long stretches of ssDNA and fork collision with the transcription machinery. Similarly, the mechanism for rare Rad51-independent gene conversions is unclear. Notably, the majority of spontaneous and damage-induced recombination events in yeast are dependent on Rad52, suggesting that Rad52's strand annealing function plays a critical role[25].

## Methods

**Yeast strains**. All yeast strains are derived from W303, corrected for the rad5-535 mutation, and are ade2::hisG (Supplementary Table 2). The ade2 inverted-repeat recombination reporter was described previously[26]. In this study, the reporter was amplified by 2-rounds PCR from the strain 2002-9D[29] and integrated at the his2 locus on chromosome 6.

The 14 TerB repeats were amplified by PCR from plasmids pNBL63 (blocking orientation) and pNBL55 (permissive orientation) and integrated 170 and 120 bp distal to the ade2Δ5' repeat, respectively. The $P_{GAL1}$-HA-Tus cassette was cloned from plasmid p415- $P_{GAL1}$-HA-Tus[30] into pRG205MX (Supplementary Reference 1), adjacent to the yeast LEU2MX selectable marker, and integrated at the LEU2 locus.

All mutant strains were constructed by genetic crosses using haploid strains in the laboratory collection, with the exception of csm2Δ, mph1Δ, and mus81Δ strains that were obtained by transformation with a KanMX ORF replacement cassette. Strains used for 2D gels additionally contained a bar1::HphMX allele generated by one-step gene replacement or by a genetic cross.

**Measurement of recombination frequency**. The percentage of Ade+ recombinants, which corresponds to recombination frequency, was measured as follows. Strains were grown for 3 days on YPAD (1% yeast extract, 2% bacto-peptone, 2% dextrose, 10 mg/L adenine) or 4 days on YPAG (1% yeast extract, 2% bacto-peptone, 2% galactose, 10 mg/L adenine) plates. Colonies of similar size (described below as initial colonies) were suspended in 1 mL water to an $OD_{600}$ close to 0.3. Cells were serially diluted and plated on YPAD or synthetic complete-adenine (SC-Ade) medium. Colonies were counted 2 days after plating and two dilutions from each initial colony were averaged. The percent Ade+ recombinants were determined by the ratio of the number of colonies growing on SC-Ade plates and YPAD plates × 100. Each data point in the graphs shows the percentage of Ade+ recombinants measured from one initial colony. Recombination frequencies were determined using the method of the median in order to avoid the impact of "jackpot" events due to the formation of recombinants very early in the growth of a colony which can lead to a large proportion of recombinant cells generated from only one recombination event. The medians, shown on graphs as black lines, were calculated for each strain and condition from multiple independent trials and are indicated in Supplementary Table 1, as well as the number of initial colonies tested. Raw data are provided in a Source Data file.

**Distribution of Ade+ recombinants**. To ensure analysis of independent NAHR events, only one Ade+ recombinant colony from each initial colony (see above) was used. Inversions and conversions were scored by PCR using a primer annealing to the his2 sequence upstream of the ade2 reporter (Olea 10:CATAGCACACACC CACTTGC) or to the ade2-n cassette (Olea 51: GAACAGTTGGTATATTAGG AGGG), and primers of opposite orientation that anneal to the TRP1 sequence between the repeats (Olea 48:GTGGCAAGAATACCAAGAGTTCC or Olea 50 GGACCA-GAACTACCTGTG), see Fig. 1e. The number of independent recombinants tested for each strain and condition is indicated in the figures. Raw data are provided in a Source Data file. In some instances, a small proportion of Ade+ recombinants gave either no PCR amplification or both PCR products. They might be due to the presence of complex rearrangements as previously reported[29]. Those were not analyzed further and are indicated as nd, for not determined, in distribution graphs.

**Statistical analysis**. Ade+ recombination frequencies were analyzed on log-transformed values by one-way Anova with a Bonferroni post-test. Spontaneous and Tus/Ter associated data were analyzed separately. Distributions of inversions and conversions among Ade+ recombinants were analyzed by a two-tailed Chi-square test. Stars indicate a significant difference with the WT strain in the same condition: *p-value < 0.05, **p-value < 0.005, ***p-value < 0.001, ****p-value < 0.0001. Exact p-values are provided in Supplementary Data 1. When relevant, exact p-values are also indicated on figures.

**Two-dimensional (2D) gel analysis of replication intermediates**. Yeast cultures were grown overnight in YEPL (1% yeast extract, 2% bacto-peptone, 10 mg/L adenine, 3% sodium DL-lactate) medium to $OD_{600} = 0.8$. Cultures were

synchronized in G1 with 1.5 µg/mL alpha factor mating pheromone (GenScript) for 3 h at 30 °C. Tus expression was induced by adding 2% Galactose (final w/v) for the final 2.5 h of the G1-arrest. Cells were released from G1-arrest by centrifugation, washing, and resuspension in a warm YEPL medium containing 100 µg/mL pronase. Arrest and release of the cultures were checked by flow cytometry. Cells were incubated for 50 min at 30 °C, then cultures were placed on ice and treated with 0.1% sodium azide to stop metabolism. The hexadecyltrimethylammonium bromide (CTAB) protocol was followed for the extraction of total genomic DNA (Supplementary Reference 2). A Qubit Flex fluorometer (Invitrogen) was used for quantification and the DNA yield was about 30 µg from each 200 mL overnight culture.

For each 2D gel, 15 µg of genomic DNA was digested overnight with 90 U ClaI. Samples were run on the first-dimension gel (0.35% agarose, 1× Tris-Borate-EDTA) at a constant voltage of 1 V/cm for ~19 h, and then stained with 0.3 µg/mL ethidium bromide. Gel strips were excised under a UV trans-illuminator, rotated by 90° and run on a second gel (1.15% agarose, 1× Tris-Borate-EDTA, 0.3 µg/mL ethidium bromide) at 4 V/cm for ~6 h at 4 °C.

**Southern blotting**. After denaturation and neutralization of the gels, DNA was transferred in 2 x SSC to positively charged nylon membranes (GE Healthcare Amersham Hybond-N+) and was then immobilized by ultraviolet cross-linking (1200J). DNA fragments were detected using a mix of five probes labeled by PCR amplification with $^{32}P$ dCTP (Perkin Elmer) described in Supplementary Fig. 1. ULTRA-hyb Ultrasensitive hybridization buffer (Invitrogen) was used for hybridization of the probes at 42 °C. Membranes were washed as recommended by the manufacturer. 2D gels were exposed for 4 h in a phosphor screen cassette and the signal was detected with a Typhoon Trio phosphoimager (GE healthcare).

**Reporting summary**. Further information on research design is available in the Nature Research Reporting Summary linked to this article.

## Data availability

All data generated or analyzed during this study are included in this published article and its Supplementary Information file, and are available from the corresponding author upon reasonable request. Source data are provided with this paper.

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

## Acknowledgements

We thank H. Mankouri and R. Rothstein for generous gifts of plasmids and strains, and W.K. Holloman, R. Reid, and members of the Symington lab for comments on the manuscript. This work was supported by grants from the National Institutes of Health (R21 ES030447, R35 GM126997, and P01 CA174653 to L.S.S.).

## Author contributions

L.M. performed all experiments. L.M. and L.S.S. contributed to the study design, data analysis, and manuscript preparation.

## Competing interests

The authors declare no competing interests.
