## [Peer Review File · Nature Communications]

Mechanism for inverted-repeat recombination induced by a replication fork barrierReviewers' Comments:

Reviewer #1:

Remarks to the Author:

In the manuscript by Marie and Symington, the investigators determine the mechanism of how inverted repeats are repaired in a replicative context by using a Tus/Ter based system that they created. They found that recombination between inverted repeats uses a unique mechanism involving Rad51, Rad52, and/or Rad59 depending upon whether gene conversion or inversions occur. The investigators later went on to genetically dissect the molecular requirements including the role of Rad51 strand invasion activity, Rad52 single-strand annealing function, the role of the Rad51 mediators, the helicases that promote fork reversal, as well as the nucleases needed to create a ssDNA template for Rad51 to load, etc. The innovation of this work is demonstrating that replication stalling at repetitive sequences uses an HR-associated mechanism that is genetically distinct from spontaneous HR and PRR-mediated template switching. This eloquent study does an outstanding job of defining this pathway and its distinct requirements using a modified Tus/Ter system in budding yeast.

Minor comments:

- 1) Since Sgs1 has multiple functions outside of DNA end resection, could *dna2Δ* cells also reduce % Ade⁺ recombinants? Perhaps a discussion or analysis of this mutant would be insightful.
- 2) Lines 213-218 refer to Figure 3C and 3D but should refer to Figure 3B.

Reviewer #2:

Remarks to the Author:

This paper uses a genetically marked construct to detect homologous recombination events between inverted repeat sequences in budding yeast. The construct also contains an array of *ter* binding sites that bind the protein Tus when it is expressed. Binding of Tus to *ter* sites blocks replication forks, allowing the authors to look specifically at recombination events caused by a protein barrier to replication fork progression. This system was used to show that inverted repeat recombination events are indeed stimulated by a protein barrier. Most recombinants formed because of the Tus block are formed by what appears to be a novel pathway compared to the pathways responsible for most spontaneous events detected with the same recombination reporter construct. The novel aspect of the new pathway is that both the strand exchange protein Rad51 and the annealing protein Rad59 are required to form viable recombinants, whereas only one of these two activities is sufficient for most spontaneous events. The novel pathway was further characterized using mutants that block various stages in homologous recombination (HR) and/or post replicative repair (PRR). This led to a hypothetical model for the underlying molecular events. This paper will likely be of interest to the readers of Nat. Comm. who study recombination and replication. The experiments are done using appropriate methods and the description of the methodology is sufficient to allow repetition. Overall, the conclusions are supported by the data. In a few instances either an alternative interpretation should be mentioned, or additional discussion/clarification would be helpful.

I have some suggestions for improving the manuscript below. They are not in order of importance. I have indicated the most important comments with an asterisk.

*I think the title should be changed to reflect that only HR involving an inversion was looked at. Something like "Homologous recombination between inverted repeats is induced by a replication fork barrier via a pathway involving both strand invasion and annealing" or something of the sort.

146. What is the "small" proportion of recombinants not analyzed further?

148. what are the #'s for the "moderate" induction of NAHR via HU?

*150. I do not understand this explanation for the lack of an effect of HU. If the amount of HU used is compatible with growth, clearly replication is not completely blocked. Perhaps the authors could consider an experiment using a higher level of HU that completely halts replication for a set period, and then allow the cells to resume growth in medium without HU. Will fork stalling by this method induce recombinants? If so, does this type of recombination resemble that induced by Tus blockage in that it depends on both RAD51 and RAD59? This approach would make it possible to determine if two different types of replication blockage involve the new pathway, and therefore show more conclusively that the new pathway is not specific to use of a protein barrier. This in turn will provide a more solid conclusion that spontaneous inverted repeat recombination does not involve replication blockage.

168. For spontaneous NAHR, the Rad51 dependent and the Rad59 dependent paths are enriched for conversions and inversions respectively, but do not display complete separation of these two outcomes. Is there a model that accounts for Rad51-independent conversions? Perhaps this deserves some mention. In addition, given the requirements of the tus/ter induced pathway, it would be of interest to know the upper limit for the fraction of spontaneous events that could be accounted for by the mechanism revealed by the tus/ter block.

*261. Regarding the modest impact of the rad5 mutation ... the papers cited, that argue for the involvement of Rad5 in reversing forks, involve only very simple biochemical experiments. Such experiments can give misleading results. I don't happen to know if there is specific in vivo evidence (e.g. EM or 2D gel) that rad5's functions include catalyzing fork regression. If there is, that evidence should be cited. If not, I think it is possible that the data presented here reflects a lack of involvement of Rad5 in fork regression in vivo. It is not worth considering the possibility that the impact of the rad5 in the mph1 mutant results from the mph1 block shunting of repair intermediates to a non-fork regression prr path that requires Rad5?

310. How much was recombination reduced in the rad1 mutant?

415. The logic that the defect of a rad5 mph1 double mutant argues against a role for Rad54 in fork regression is unclear to me. Why does that result exclude the possibility that Rad54 is cooperating with Rad5 and Mph1 to reverse forks?

477. Perhaps there should be some mention of how the method of using the median from a large # of independent frequency measurement reduces the impact of jackpots and obviates the need for more traditional calculation of mutation rates.

*Figure 6. I appreciate fork regression figures are difficult to draw, but I suspect the version presented will be incomprehensible to almost all readers, even those with strong backgrounds in recombination mechanism. For example, the specific structures that are proposed to form during steps 5 and 7 are quite obscure. It might be helpful to not use overlaid boxes that obscure the two strands of the duplex to represent the different regions. The reader is going to want to see where and how the D-loops are and how they are oriented as well as where the heteroduplex DNA is. Pairs of ½ arrowheads and color coding of both strands could be used to indicate the position and orientation of the repeated regions. Note that is this approach is used, it would be necessary to use two different arrowhead styles for indicating the orientation of repeats and for indicating 3' strand ends.

Reviewer #3:
Remarks to the Author:

In this manuscript, the Marie and Symington address the effect of a defined replication-fork block mediated by Tus/Ter on recombination between *ade2* inverted repeats in budding yeast. Two types of outcomes were measured for spontaneous events and those associated with the block: conversion of the full-length repeat to WT and conversion associated with inversion of the segment between the repeats. The authors demonstrate that the genetic requirements for Tus/Ter events are distinctly different from those of either spontaneous HR or post-replication repair. Based on these requirements, a model is proposed that incorporates fork reversal triggered by the block and subsequent invasion/interaction of repeats on opposite sides of the fork. This is a well-done study; the data are convincing and the manuscript was very clear and a pleasure to read. Importantly, this study expands possible mechanisms for structural variations associated with genomic disorders. Comments that should be considered when revising the manuscript are below.

1. Lines 167-8: statistical analysis (contingency Chi square) of the distribution changes should be done. Mutants are different from each other and from WT. Same comment applies to lines 177-178.
2. Line 228 – it would help to remind the reader here (rather than later) that the reduction in the *rad57* background was much greater than in the *rad51*.
3. Don't the different genetic requirements of Tus/Ter and spontaneous events suggest that most spontaneous events have a different origin?
4. There are a few changes to Fig 6 that would make things clearer for the reader:
 - Indicate which *ade2* repeat is full length and which is truncated, which will make it clear that the full length eventually invades the truncated copy.
 - In (4), why are both ends shown as being degraded? Wouldn't only the 5' end be degraded? The requirement of Rad1 would be consistent with clipping of a 3' unpaired tail after invasion of the truncated copy.
 - Going from (6) to (7) shows fork reversal and replication. Reversal only involves the truncated copy.
 - The unpaired heterology in (7) right is not clear and the strands should be clearly unpaired.
5. A prediction from the model proposed is that the longer tract synthesis required for inversion should be impaired more in the *pol32* mutant than conversion. If this was not examined, it should be.
6. The authors might want to consider replacing Fig 4D (it is repeated in the more complete Fig 6 model) with Fig 5, which would reduce the number of figures.
7. The white sectoring in Fig S2 is much clearer than in Fig 1B. Please use a better picture in the main text.
8. Remove/replace "remarkably" in the abstract
9. Avoid introducing acronyms that are used only once (e.g., LCR, CNV in Introduction)

We thank the reviewers for their positive comments, and suggestions for text modifications and additional experiments to support the conclusions of the study. Below is a point-by-point response to the reviewers' comments.

Reviewer #1:

In the manuscript by Marie and Symington, the investigators determine the mechanism of how inverted repeats are repaired in a replicative context by using a Tus/Ter based system that they created. They found that recombination between inverted repeats uses a unique mechanism involving Rad51, Rad52, and/or Rad59 depending upon whether gene conversion or inversions occur. The investigators later went on to genetically dissect the molecular requirements including the role of Rad51 strand invasion activity, Rad52 single-strand annealing function, the role of the Rad51 mediators, the helicases that promote fork reversal, as well as the nucleases needed to create a ssDNA template for Rad51 to load, etc. The innovation of this work is demonstrating that replication stalling at repetitive sequences uses an HR-associated mechanism that is genetically distinct from spontaneous HR and PRR-mediated template switching. This eloquent study does an outstanding job of defining this pathway and its distinct requirements using a modified Tus/Ter system in budding yeast.

Minor comments:

1) Since Sgs1 has multiple functions outside of DNA end resection, could *dna2Δ* cells also reduce % Ade⁺ recombinants? Perhaps a discussion or analysis of this mutant would be insightful.

Response: The deletion of *DNA2* is lethal in *S. cerevisiae* but can be rescued by deletion of *PIF1* or use of the *pif1-m2* mutant allele. While loss of *PIF1* impairs mitochondrial function, the *pif1-m2* allele reduces only nuclear Pif1 and avoids growth defects associated with *PIF1* deletion. We introduced the *pif1-m2* and *dna2Δ pif1-m2* double mutations in the strain containing the *ade2* reporter and 14 *Ter* repeats in the blocking orientation. The results are presented lines 301-310 and Figure 4b. In the absence of Tus, we did not detect any difference in the frequency of spontaneous recombination between the WT and the mutant strains. When Tus was expressed, the *pif1-m2* mutant strain showed an increased frequency of recombination comparable to the WT strain. However, we observed a 3.4-fold decreased frequency of Tus-induced recombination in the *dna2Δ pif1-m2* double mutant compared to the *pif1-m2* strain. These results are consistent with a role for Sgs1-Dna2-dependent long-range resection in replication-associated NAHR. The potential role of Sgs1-Dna2 is now also mentioned in the discussion line 425 and in the model (Figure 5).

2) Lines 213-218 refer to Figure 3C and 3D but should refer to Figure 3B.

Response: This mistake has been corrected in the text.

Reviewer #2:

This paper uses a genetically marked construct to detect homologous recombination events between inverted repeat sequences in budding yeast. The construct also contains an array of *ter* binding sites that bind the protein Tus when it is expressed. Binding of Tus to *ter* sites blocks replication forks, allowing the authors to look specifically at recombination events caused by a protein barrier to replication fork progression. This system was used to show that inverted repeat recombination events are indeed stimulated by a protein barrier. Most recombinants formed because of the Tus block are formed by what appears to be a novel pathway compared to the pathways responsible for

most spontaneous events detected with the same recombination reporter construct. The novel aspect of the new pathway is that both the strand exchange protein Rad51 and the annealing protein Rad59 are required to form viable recombinants, whereas only one of these two activities is sufficient for most spontaneous events. The novel pathway was further characterized using mutants that block various stages in homologous recombination (HR) and/or post replicative repair (PRR). This led to a hypothetical model for the underlying molecular events. This paper will likely be of interest to the readers of Nat. Comm. who study recombination and replication. The experiments are done using appropriate methods and the description of the methodology is sufficient to allow repetition. Overall, the conclusions are supported by the data. In a few instances either an alternative interpretation should be mentioned, or additional discussion/clarification would be helpful.

I have some suggestions for improving the manuscript below. They are not in order of importance. I have indicated the most important comments with an asterisk.

1) *I think the title should be changed to reflect that only HR involving an inversion was looked at. Something like “Homologous recombination between inverted repeats is induced by a replication fork barrier via a pathway involving both strand invasion and annealing” or something of the sort.

Response: We had to shorten the length of the title to conform to the 15-word limit for Nature Communications. The new title is “Mechanism for inverted-repeat recombination induced by a replication fork barrier”.

2) Line 146. What is the “small” proportion of recombinants not analyzed further?

Response: The recombinants whose structure that could not be determined by the PCR method represent 2.4% of the tested recombinants in presence of CPT. This number is now indicated in the text line 147.

We think these recombinants might be due to more complex rearrangements, such as triplications, as reported in a previous study. This is now mentioned in the methods part called “Distribution of Ade⁺ recombinants” lines 520-522.

3) Line 148. What are the #'s for the “moderate” induction of NAHR via HU?

Response: On plates, HU induces recombination in the WT strain 2.4-fold. This number is now indicated in the text line 148.

4) * Line 150. I do not understand this explanation for the lack of an effect of HU. If the amount of HU used is compatible with growth, clearly replication is not completely blocked. Perhaps the authors could consider an experiment using a higher level of HU that completely halts replication for a set period, and then allow the cells to resume growth in medium without HU. Will fork stalling by this method induce recombinants? If so, does this type of recombination resemble that induced by Tus blockage in that it depends on both RAD51 and RAD59? This approach would make it possible to determine if two different types of replication blockage involve the new pathway, and therefore show more conclusively that the new pathway is not specific to use of a protein barrier. This in turn will provide a more solid conclusion that spontaneous inverted repeat recombination does not involve replication blockage.

Response: We have removed the unsatisfactory explanation for the lack of a strong effect of HU. As suggested, we checked if higher concentrations of HU would induce Ade⁺ recombination in liquid cultures. The results are presented in the text, lines 150-152, and in Supplementary Fig 2d. WT cells were grown to OD~0.3 in YPAD then exposed to 0.02% MMS, 200mM HU or 400mM HU for 3h. Cells were washed and allowed to resume growth for 1.5h before being plated on YPAD and SC-Ade plates. Colonies were counted after 2 days and recombination frequencies are reported in

Supplementary Fig 2d,f. A very slight increase of recombination was observed in presence of high concentrations of HU, consistent with the moderate induction we also detected using low HU concentration in solid medium (Supplementary Fig 2b). This observation is also consistent with previous work using a direct repeat recombination assay in *S. pombe* (Ahn, Osman, and Whitby 2005). Considering that the difference with the no drug condition was not statistically significant, we did not test the requirement for Rad51 and Rad59.

5) Line 168. For spontaneous NAHR, the Rad51 dependent and the Rad59 dependent paths are enriched for conversions and inversions respectively, but do not display complete separation of these two outcomes. Is there a model that accounts for Rad51-independent conversions? Perhaps this deserves some mention. In addition, given the requirements of the *tus/ter* induced pathway, it would be of interest to know the upper limit for the fraction of spontaneous events that could be accounted for by the mechanism revealed by the *tus/ter* block.

Response: Rad51-independent recombination has been noted in most yeast recombination assays, particularly for spontaneous events. Because all these events require Rad52, it is assumed that the Rad52 strand annealing activity is important to promote them. This is now mentioned in the discussion lines 476-478.

Spontaneous NAHR events occur at a frequency of 0.615% in the WT strain and the numbers drop to 0.39% in the *rad51* strain (no Rad51-mediated events, no *Tus/Ter*-like events, only Rad59-mediated events), 0.46% in the *rad59* strain (no Rad59-mediated events, no *Tus/Ter*-like events, only Rad51-mediated events) and <0.01% in the double mutant (see Supplementary Table 1). Theoretically, the total frequency of spontaneous events should be the addition of Rad51-mediated events, Rad59-mediated events and *Tus/Ter*-like events. However the sum of only Rad51-mediated events (0.46%) and only Rad59-mediated events (0.39%) is already higher than the spontaneous frequency we measured in the WT (0.615%). We think this might be due to a change in the number of initiating lesions in the mutants compared to the WT strain. In this condition, it seems complicated to deduce a percentage of spontaneous events that could be accounted for by the mechanism revealed by *Tus/Ter* induction.

6) * Line 261. Regarding the modest impact of the *rad5* mutation ... the papers cited, that argue for the involvement of Rad5 in reversing forks, involve only very simple biochemical experiments. Such experiments can give misleading results. I don't happen to know if there is specific *in vivo* evidence (e.g. EM or 2D gel) that *rad5*'s functions include catalyzing fork regression. If there is, that evidence should be cited. If not, I think it is possible that the data presented here reflects a lack of involvement of Rad5 in fork regression *in vivo*. It is not worth considering the possibility that the impact of the *rad5* in the *mph1* mutant results from the *mph1* block shunting of repair intermediates to a non-fork regression *pr* path that requires Rad5?

Response: In addition to the *in vitro* fork regression data for Rad5, Minca and Kowalski 2010 and Bryant et al. 2019 provide evidence by 2D gels suggesting an effect of Rad5 on fork regression *in vivo*. Furthermore, Xue et al. 2014, showed that the *mph1-ΔS1* mutant allele, whose fork regression activity was not inhibited by Smc5/6, can partially suppress the *rad5-DEAA* helicase dead mutant MMS sensitivity, but not the *mms2Δ* mutant sensitivity, suggesting a specific compensation for the *rad5* mutant's fork regression defect. Even though more direct data such as EM are still pending, we believe that these studies support a role for Rad5 in fork regression. These references have been added to the manuscript (line 261). We also added a sentence of the possible role for Rad5 in a recombination mechanism that occurs in the absence of fork regression (lines 274-276).

7) Line 310. How much was recombination reduced in the *rad1* mutant?

Response: *Tus/Ter* induced recombination was reduced 8-fold in the *rad1* mutant compared to the WT strain. This number is now indicated in the text line 318.

8) Line 415. The logic that the defect of a *rad5 mph1* double mutant argues against a role for Rad54 in fork regression is unclear to me. Why does that result exclude the possibility that Rad54 is cooperating with Rad5 and Mph1 to reverse forks?

Response: We mean that Rad54 is probably not the major mediator of fork reversal, otherwise we would not expect the double mutant *rad5Δ mph1Δ* to have such a strong defect in Tus/Ter induced recombination. However we agree this result should not exclude the possibility that Rad54 cooperates with Rad5 and Mph1, as we suggest lines 416-417 and 421.

9) Line 477. Perhaps there should be some mention of how the method of using the median from a large # of independent frequency measurement reduces the impact of jackpots and obviates the need for more traditional calculation of mutation rates.

Response: This complementary information has been added to the paragraph called “measurements of recombination frequency” in the methods part line 506-509.

10) * Figure 6. I appreciate fork regression figures are difficult to draw, but I suspect the version presented will be incomprehensible to almost all readers, even those with strong backgrounds in recombination mechanism. For example, the specific structures that are proposed to form during steps 5 and 7 are quite obscure. It might be helpful to not use overlaid boxes that obscure the two strands of the duplex to represent the different regions. The reader is going to want to see where and how the D-loops are and how they are oriented as well as where the heteroduplex DNA is. Pairs of ½ arrowheads and color coding of both strands could be used to indicate the position and orientation of the repeated regions. Note that if this approach is used, it would be necessary to use two different arrowhead styles for indicating the orientation of repeats and for indicating 3' strand ends.

Response: In an effort to make the model more comprehensible, we have added 2 zoom boxes, called 5.1 and 5.2, to better depict the molecular structures that are proposed to form during the strand invasion at step 5. We also added arrowheads to all DNA strands on the model to make the orientations of the strands clearer. Finally, we have separated step 7 into two parts called step 7-fork reversal and step 8-replication so the structures and processes that are proposed to be involved at these steps would be easier to visualize.

Reviewer #3:

In this manuscript, the Marie and Symington address the effect of a defined replication-fork block mediated by Tus/Ter on recombination between *ade2* inverted repeats in budding yeast. Two types of outcomes were measured for spontaneous events and those associated with the block: conversion of the full-length repeat to WT and conversion associated with inversion of the segment between the repeats. The authors demonstrate that the genetic requirements for Tus/Ter events are distinctly different from those of either spontaneous HR or post-replication repair. Based on these requirements, a model is proposed that incorporates fork reversal triggered by the block and subsequent invasion/interaction of repeats on opposite sides of the fork. This is a well-done study; the data are convincing and the manuscript was very clear and a pleasure to read. Importantly, this study expands possible mechanisms for structural variations associated with genomic disorders. Comments that should be considered when revising the manuscript are below.

1) Lines 167-8: statistical analysis (contingency Chi square) of the distribution changes should be done. Mutants are different from each other and from WT. Same comment applies to lines 177-178.

Response: The Chi-square statistical analysis of the distribution changes between the *rad51Δ* and *rad59Δ* mutants in spontaneous condition has been done and is now indicated by 4 stars on Figure 2b.

The Chi-square analysis was also done for the Tus/*Ter* induced condition reported in Figure 2e but revealed no statistically significant difference between the two mutants or with the WT strain, thus no stars are indicated on the graph.

2) Line 228 – it would help to remind the reader here (rather than later) that the reduction in the *rad57* background was much greater than in the *rad51*.

Response: This information has been added line 225 as suggested by the reviewer.

3) Don't the different genetic requirements of Tus/*Ter* and spontaneous events suggest that most spontaneous events have a different origin?

Response: We agree it probably does and comment on this at the end of the Discussion. It would be interesting to know how many of the spontaneous events are mediated by the Rad51-Rad59-Rad52 pathway revealed by Tus/*Ter*, however this seems complicated to estimate. We address this point with more details in our response to comment #5 from reviewer 2.

4) There are a few changes to Fig 6 that would make things clearer for the reader:

- Indicate which *ade2* repeat is full length and which is truncated, which will make it clear that the full length eventually invades the truncated copy.

Response: We have added the frameshift and the truncation to the model presented in Figure 5 (previously Figure 6) so the invasion of the truncated copy by the full length copy is clearer to visualize, as well as the repair of the frameshift mutation. In the alternative model presented in Supplementary Fig 4 the frameshift and truncation are also shown.

- In (4), why are both ends shown as being degraded? Wouldn't only the 5' end be degraded? The requirement of Rad1 would be consistent with clipping of a 3' unpaired tail after invasion of the truncated copy.

Response: In our reporter system, fork reversal needs to regress several kb from the Tus/*Ter* barrier to reach the full length *ade2* copy. After resection of the 5'-terminated strand by Mre11 and Exo1/Sgs1-Dna2, the 3' overhang would be very long and encompass most of the *ade2* reporter. These sequences would need to be cleaved so the invading DNA contains the sequence downstream from the position of the frameshift. This cleavage could occur by Rad1-Rad10 clipping after invasion of the truncated copy or before invasion when the long ssDNA is exposed at the regressed fork. These two possibilities are now mentioned in the discussion lines 426-428.

- Going from (6) to (7) shows fork reversal and replication. Reversal only involves the truncated copy.

Response: This is correct. We have added 2 new steps in the model representing fork reversal (7) and replication (8) separately.

- The unpaired heterology in (7) right is not clear and the strands should be clearly unpaired.

Response: We modified the design of Figure 5 and Supplementary Fig 4 so it would be clearer that the strands are not paired along the intervening sequence and that instead they form a heterologous loop.

5) A prediction from the model proposed is that the longer tract synthesis required for inversion should be impaired more in the *pol32* mutant than conversion. If this was not examined, it should be.

Response: We agree this is a reasonable prediction. We tested 42 independent Tus-induced recombinants in the *pol32Δ* strain and found the distribution of events was not significantly different from the WT strain (lines 344-345 and Supplementary Fig 3c). An alternative model for the late stages of NAHR, taking this result into account, is presented in Supplementary Fig 4 and discussed lines 450-465. We did not replace the original model with the alternative model as we think it is more complex and also presents its own caveat, which we discuss line 462-465.

6) The authors might want to consider replacing Fig 4D (it is repeated in the more complete Fig 6 model) with Fig 5, which would reduce the number of figures.

Response: As suggested, the *pol32Δ* data have been moved to Figure 4d and the repeating model has been removed.

7) The white sectoring in Fig S2 is much clearer than in Fig 1B. Please use a better picture in the main text.

Response: A better picture with clearer white sectoring is now presented in Figure 1b.

8. Remove/replace "remarkably" in the abstract

Response: "Remarkably" has been removed from the abstract.

9. Avoid introducing acronyms that are used only once (e.g., LCR, CNV in Introduction)

Response: We have removed all the unnecessary acronyms.

Ahn, J. S., F. Osman, and M. C. Whitby. 2005. "Replication fork blockage by RTS1 at an ectopic site promotes recombination in fission yeast." *EMBO J* 24 (11):2011-23. doi: 10.1038/sj.emboj.7600670.

Bryant, E. E., I. Šunjevarić, L. Berchowitz, R. Rothstein, and R. J. D. Reid. 2019. "Rad5 dysregulation drives hyperactive recombination at replication forks resulting in cisplatin sensitivity and genome instability." *Nucleic Acids Res* 47 (17):9144-9159. doi: 10.1093/nar/gkz631.

Minca, E. C., and D. Kowalski. 2010. "Multiple Rad5 activities mediate sister chromatid recombination to bypass DNA damage at stalled replication forks." *Mol Cell* 38 (5):649-61. doi: 10.1016/j.molcel.2010.03.020.

Xue, X., K. Choi, J. N. Bonner, T. Chiba, Y. Kwon, Y. Xu, H. Sanchez, C. Wyman, H. Niu, X. Zhao, and P. Sung. 2014. "Restriction of replication fork regression activities by a conserved SMC complex." *Mol Cell* 56 (3):436-445. doi: 10.1016/j.molcel.2014.09.013.

Reviewers' Comments:

Reviewer #2:

Remarks to the Author:

The revisions provided by the author are satisfactory to me and I recommend publication.

Reviewer #3:

Remarks to the Author:

The authors have nicely addressed the comments of reviewers and II have no further suggestions.